# The Assessment of the Transversal Rupture Strength (TRS) and Hardness of WC-Co Specimens Made via Additive Manufacturing and Sinter-HIP

**Ovidiu-Darius Jucan [1,*], Rareş-Vasile Gădălean [2,3], Horea-Florin Chicinaş [1,2], Nicolae Bâlc [3] and Cătălin-Ovidiu Popa [1]**

[1] Materials Science and Engineering Department, Technical University of Cluj-Napoca, 103-105, Muncii Avenue, 400641 Cluj-Napoca, Romania; horea.chicinas@stm.utcluj.ro (H.-F.C.); catalin.popa@stm.utcluj.ro (C.-O.P.)

[2] Gühring Romania, Constructorilor Street 32, 407035 Apahida, Romania; rares.gadalean@guehring.de

[3] Department of Manufacturing Engineering, Technical University of Cluj-Napoca, 103-105, Muncii Avenue, 400641 Cluj-Napoca, Romania; nicolae.balc@tcm.utcluj.ro

\* Correspondence: darius.jucan@stm.utcluj.ro

**Abstract:** This study is focused on the mechanical properties of WC-Co composites obtained via Selective Laser Sintering (SLS) using PA12 as a binder. The as-printed samples were thermally debonded and sintered, first in a vacuum, and then sinter-HIP (Hot Isostatic Pressure) at 1400 °C, using 50 bar Ar, which has led to relative densities up to 66%. Optical metallographic images show a microstructure consisting of WC, with an average grain size in the range of 1.4–2.0 μm, with isolated large grains, in a well-distributed Co matrix. The shrinkage of the samples was 43%, with no significant shape distortion. The printing direction of the samples significantly impacts the transversal rupture strength (TRS). Nevertheless, the mechanical strength was low, with a maximum of 612 MPa. SEM images of the fracture surface of TRS samples show the presence of defects that constitute the cause of the low measured values. The hardness values position the obtained composites in the medium coarse classical cemented carbides range. The results were also related to the amount of free Co after sintering, close to the initial one, as assessed by magnetic measurements, indicating a low degree of interaction with PA12 decomposition products.

**Keywords:** WC-Co composite; additive manufacturing; transversal rupture strength

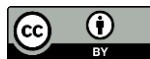

## 1. Introduction

The WC-Co composites, also known as cemented carbides, are a class of materials well-known for their excellent combination of hardness and toughness. Parts made of WC-Co are generally used in industries like, mining, machining, cold forming, etc., where the wear properties play a significant role [1,2]. The mechanical properties of cemented carbides are influenced mainly by the mean grain size of WC, which is in size range of 0.3–7.0 μm, and the amount of Co, ranging from 3.0 to 30.0% [3]. Several pioneering studies concerning the mechanical properties of WC-Co theoretically and empirically explain the factors that influence the mechanical behaviour of cemented carbides. A relationship between the hardness of WC-Co and the mean distance between carbide grains was found to be viable, which can be related to its microstructure [4]. In ultra-fine cemented carbide grade, where WC grain size is <0.5 μm, the parts exhibit high hardness, thus, high wear resistance due to preventing abrasive materials from penetrating in depth.

The manufacturing route of cemented carbides is carried out by Powder Metallurgy methods, where the raw materials are first homogenised, then pressed depending on the geometrical complexity, followed by green machining and liquid phase sintering. Often,

the green body is pre-sintered to increase the strength to withstand the machining operation, which intends to impose the geometrical features. The traditional manufacturing process of cemented carbide parts involves high costs generated by the machining of green or pre-sintered parts in the case of high geometrical complexity parts and also by using dies which have intricated shapes and are costly to produce.

Additive manufacturing (AM) promises to overcome the limitations imposed by traditional manufacturing by eliminating the need for tooling and other manufacturing operations, overall reducing manufacturing costs.

Until today, a few studies have been reported concerning AM of cemented carbides. Selective Laser Melting (SLM) sits at the top of the AM techniques used in WC-Co processing. In the case of cemented carbides, the processes that develop high temperatures during the layer-by-layer processing lead to decarburisation, chemical imbalances and Co evaporation [5]. During the repeated heating-cooling cycles of layer-by-layer building up, the Co behaviour differs from the traditional sintering process. Moreover, WC decomposes when exposed to high temperatures [6].

Binder Jetting (BJ) is a relatively new AM process, which employs a binder to selectively glue areas from the powder bed; thus, the printing process results in a green body that requires further processing to increase the density. Using BJ, K. Enneti [7] reached a near theoretical density for a WC-12Co powder, using a 6.25 $g/cm^3$ apparent density raw powder and sintered at 1485 °C/13 bars Ar pressure. The mechanical properties and microstructural results were similar to the WC-12Co medium grain size parts produced by the traditional manufacturing route. Furthermore, it has been proven that WC-Co produced by BJ has appropriate mechanical properties, which confer to BJ a high potential in the manufacturing of WC-Co [8].

In contrast to BJ and SLM, Indirect Selective Laser Sintering (SLS) uses a low-power laser to selectively melt areas from the powder bed, consequently creating necks between the particles. In the SLS of WC-Co, a mixture of cemented carbide powder and a polymer binder is first printed layer by layer using a standard 3D printer. The printed object is then subjected to a heat treatment process that removes the polymer binder, leaving behind a "green" cemented carbide part that is still porous and fragile. This green part is sintered in a high-temperature furnace to create a final dense and strong cemented carbide part. The main advantage of using SLS in printing cemented carbide parts is that no support structures are needed due to the surrounding powder from the powder bed, which supports the printed parts. Secondly, it allows freedom of design since the parts are produced in a layer-by-layer fashion, where complex structures can be printed with intricated cooling channels. The process usually occurs in $N_2$, at low temperatures, below the melting point or glass transition temperature of the organic binder. The SLS was employed in processing metallic and ceramic materials together with an organic binder [9–14]. The purpose of the organic binder is to melt under the laser beam to create bonds between the particles. In this way, the parts resulting from the printing process are in a green state. Therefore, the printed parts undergo no deleterious effects specific to using high processing temperatures, like in the case of SLM [5]. By using SLS in printing WC-Co parts, it allows the green parts to be processed similarly to the parts that are conventionally produced. The as-printed green parts are thermally debonded to remove the organic binder used in printing. Subsequent to the debinding operation, liquid phase sintering (LPS) further desifiying the parts. During LPS, the densification mechanismslike rearrangement, solution-reprecipitation, and solid phase sintering, are active to facilitate the densification of the part. Aside from our previous work [15], based on our knowledge, no other studies have been reported of WC-Co produced via SLS and sinter-HIP using an industrial approach, where the mechanical properties of the parts are assessed. In this study, we are looking at manufacturing cemented carbide parts using a ready-to-press WC-Co powder and polyamide, both commercially available via SLS using a commercial SLS machine, followed by Sinter-HIP using an industrial furnace used in sintering of cemented carbides.

The assessment of the mechanical properties, like hardness and mechanical strength, has also been performed.

## 2. Materials and Methods

The employed cemented carbide used in this research is a commercially ready-to-press spray-dried spherical particles WC-12Co powder (GTP, Towanda, PA, USA) with 34 µm median size, used as received. The powder exhibits good flowability while its bulk density is 2.90 g/cm³, a typical powder used in the traditional manufacturing of cemented carbides. As an organic binder for building green specimens, commercial Polyamide PA12 powder (ALM, TX, USA) has been used as received. Polyamide, commonly known as nylon, is a popular material for additive manufacturing due to its unique properties. Polyamide is a thermoplastic material that can be melted and re-solidified repeatedly without any significant changes to its properties, making it suitable for use in additive manufacturing processes like selective laser sintering (SLS). The PA12 powder shows a bulk density of 0.46 g/cm³, while the average particle size is 55 µm, a well-known material used in SLS. Details regarding the WC-Co and PA12 powder properties can be found in our prior study [15]. The appropriate content of PA12 is 20 wt.%, which was empirically determined in our previous research. Figure 1 shows the morphology of the WC-12Co and PA12 powders used in this study. WC-12Co powder has spherical hollow granules with 10–74 µm diameters with a relatively porous microstructure; pores are relatively uniformly distributed within granules walls, Figure 1a. PA12 binder particles display a spherical shape; the average diameter is around 55 µm, Figure 1b. The mixture consisted of WC-12Co–20 wt. % PA12 was homogenised for 4h using a 3D industrial mixer. Cube samples of 20 × 20 mm were printed together with samples of Φ4.6 × 35 mm used for the transversal rupture strength (TRS) test. A commercial machine, Sintering 2000 (DTM Corporation/3DSystems, Santa Clarita, CA, USA), was equipped with a 50 W $CO_2$ laser to print the green samples. The printing process was performed in an $N_2$ atmosphere to avoid thermal oxidation while the parts were produced with the part bed temperature set to 170 °C.

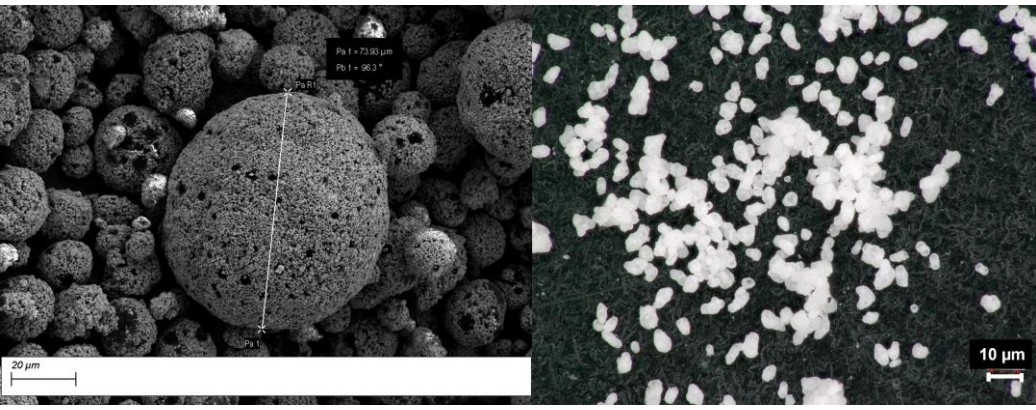

**Figure 1.** Morphology of WC-12Co powder (**a**) and PA12 powder (**b**).

The as-printed green specimens are thermally burnout at 800 °C for 4 h to remove completely the organic binder, followed by sintering in vacuum at 1400 °C for 2 h using a PVA Tepla /Pfeiffer Balzers COV 733 R industrial furnace (Westhausen, Germany). Subsequently, sintering in vacuum was performed at 1200 °C followed by HIP at 1400 °C with 50 bar Ar for 1 h, using an ALD VKP 50/50/170 industrial sinter-HIP furnace (Hanau, Germany). The dimensions of the cube shape samples were measured after each sintering cycle to calculate the shrinkage. Density determination was performed by the gravimetric method.

The free cobalt content of sintered samples was assessed by magnetic measurements using a Förster Koerzimat 1.097 HCJ unit (Reutlingen, Germany). The sintered samples were polished and etched using Murakami reagent for metallographic analysis using a

Leica DM6000 M optical microscope (Wetzlar, Germany). A Sigma Zeiss scanning electron microscope (SEM) (Jena, Germany) was also used for the green and sintered samples' microstructural characterisation. The Vickers hardness test has been performed using a FISCHERSCOPE HM2000 (Miami, FL, USA) equipment with a 0.3 kgf indentation force. The mechanical strength has been evaluated using the 3-point bending test known as transversal rupture strength, accordingly with DIN 3327 and ISO 3327, using a Zwick UPM 1475 (Ulm, Germany) with Zmart Pro machine test.

## 3. Results

### 3.1. Investigation of the Printed and Sintered WC-Co Specimens

A low powder bulk density can lead to voids, porosity, and weaker mechanical properties in the final part. A high powder bulk density can also lead to more uniform energy absorption during laser exposure, resulting in a more uniform melting and sintering process. The green samples processed by SLS showed low relative densities (around 15%) due to the low bulk density of the starting powder and the high amount of PA12 (20%), which decreases density overall. Selective laser sintering is categorised as a powder bed process. The parts are 3D printed, whereas no pressure is applied to increase the contact between particles. Therefore, the green density of the printed parts is almost equivalent to the bulk density of the used powder mixture. Due to low green density and high organic binder content (20%), the samples have undergone shrinkage up to 43% during debinding, vacuum sintering and the sinter-HIP process, exceeding the shrinkage that takes place in processes like MIM (metal injection moulding) [16] or even Binder Jetting [7]. When it comes to the sintering of cemented carbides granules, regardless of the sintering cycle, a pore-free microstructure can be obtained only for granules with a specific bulk density and particle size distribution [17]. In our case, the lack of close contact between particles, the use of hollow granules, and the high amount of organic binder hindered the specimens from attaining full densification after the sintering process. In Figure 2 is presented the contrast between cube shape samples before and after the debinding and sintering process. There is a big difference between samples before and after sintering, showing high shrinkage during the process. Concerning the sintered sample, the uneven deformation that has taken place due to the friction with the support during sintering can be noticed, a phenomenon also known as "elephant foot". The term "elephant foot" describes a condition where the bottom surface of a sintered part has a bulging or flattened appearance. In the green state, the samples contained three phases: WC-Co granules, previously molten PA12 and pores, while after sintering, the organic binder removal led to new pores formation. The densification during the debinding and liquid phase sintering process was significant with respect to the green density, indicating the removal of a high proportion of the pores, though not enough to attain the appropriate densification typically needed in the metal cutting industry.

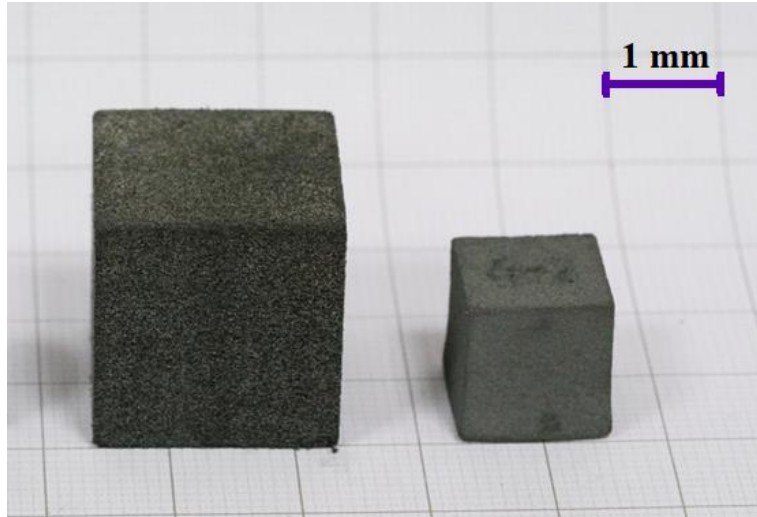

**Figure 2.** Contrast between cube shape samples before and after the debinding-sintering process, showing the shrinkage.

Samples have presented low relative densities after vacuum sintering and sinter-HIP, the WC-12Co, Table 1. The relative ineffectiveness of sinter-HIP is related to the open porosity within the samples, which hinders pore closure. In order to have effective densification during sinter-HIP, close porosity is needed, therefore, at higher levels of densification. The low densification after the sintering process is due to the low green density generated by the high amount of PA12 and the low bulk density of the starting powder. In Figure 3 is presented the microstructure after sintering of the WC-12Co specimens.

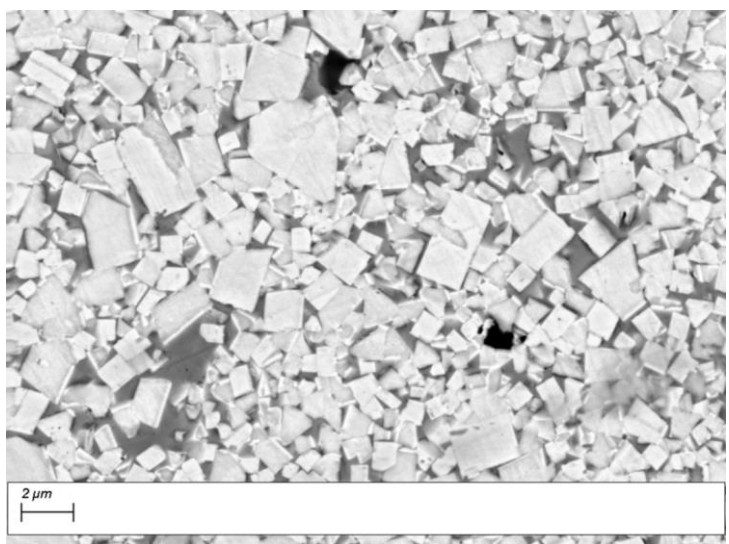

**Figure 3.** SEM image with the microstructure of sintered WC-12Co + 20% PA12.

**Table 1.** Properties of the WC-Co samples.

| $Q_{green}$ [%] | $Q_{vacuum-sintered}$ [%] | $Q_{sinter-HIP}$ [%] | $M_s$ [gsscm³/g] | $H_c$[Oe] | Shrinkage [%] | % Free Co |
|---|---|---|---|---|---|---|
| 15 | 61.3 | 66 | 217 | 148 | 43 | 10.8 |

The microstructure is similar to the medium grain size cemented carbide grade, consisting of 1.4–2.0 μm grain in a well-distributed Co matrix. As sustained by the literature data, sintering in inert gases generally acts as an inhibitor of excessive grain increase, facilitating microstructure refinement [18,19]. The most employed atmosphere for sintering

cemented carbides are argon, nitrogen and hydrogen. Using inert gases can prevent the oxidation of the metallic binder (cobalt), preventing excessive grain growth during sintering. Small and large pores can also be found that have not been removed during the sintering process, and cobalt-rich areas are considered to be former pores that have been filled with a liquid phase during sintering.

The properties of the WC-Co specimens are presented in Table 1. The magnetic measurements show a high magnetisation generated by high organic residues after PA12 decomposition. The free cobalt content in cemented carbides refers to the amount of cobalt not bound to the carbide phase and is present as metallic cobalt in the material. At the same time, the free ferromagnetic Co is 10.8%, which leads to the conclusion that a small amount of the metal binder has reacted and formed compounds during sintering due to the high amount of carbon. More details about the magnetic measurements and structural characterisation can be found in our previous work [15].

### 3.2. Mechanical Properties of the Printed and Sintered WC-Co Specimens

Transverse rupture strength, also known as the bending strength test, represents the most common way to assess the mechanical strength of cemented carbides. The specimens of a specified length, with a chamfered, rectangular cross-section, are placed on two supports and loaded centrally until the fracture occurs accordingly to the standardised DIN 3327 3-bending test method. To assess the mechanical strength via TRS, 7 (seven), measurements have been carried out for each sample printed horizontal and vertical to the printing direction. The mechanical strength of cemented carbides depends on several factors, including the carbide phase's type and amount, the carbide particles' size and distribution, and the metal binder's quality. The test involves placing a material sample across two supports and applying a load at the center of the sample using a third support. The load is gradually increased until the sample breaks. The specimens were sintered and ground before undergoing the test. The TRS has been calculated using the following equation:

$$TRS = (3FL) / (2bh^2) \tag{1}$$

where:

TRS is the transverse rupture strength.
F is the maximum applied load.
L is the distance between the support points.
b is the width of the specimen.
h is the thickness of the specimen.

In Figure 4 are presented the plotted TRS values for the WC-Co samples compared with the TRS values of cemented carbides conventionally manufactured by the Powder Metallurgy route [19]. For the WC-12Co samples built vertically during the printing process, higher mechanical strength (612 $\pm$ 66.38 MPa) has been obtained compared with the samples horizontally built (328 $\pm$ 20.38 MPa). The difference between the TRS values of the samples is related to the printing direction of the green samples during the SLS process since the obtained structure is not isotropic. Cemented carbides typically exhibit anisotropic mechanical properties due to the directional nature of the carbide particles. The orientation of the carbide particles and their distribution in the metallic binder can affect the material's mechanical properties in different directions. Regardless of the TRS values obtained for the WC-12Co processed by SLS and sinter-HIP, the transversal rupture strength is too low compared with the TRS of cemented carbides conventionally processed by Powder Metallurgy for the same WC-Co powder grade, this being highly related to the final density of the parts. According to Figure 4, the same cemented carbide grade, manufactured conventionally, shows a mechanical strength four times higher (2840 MPa) than the WC-Co processed in this study.

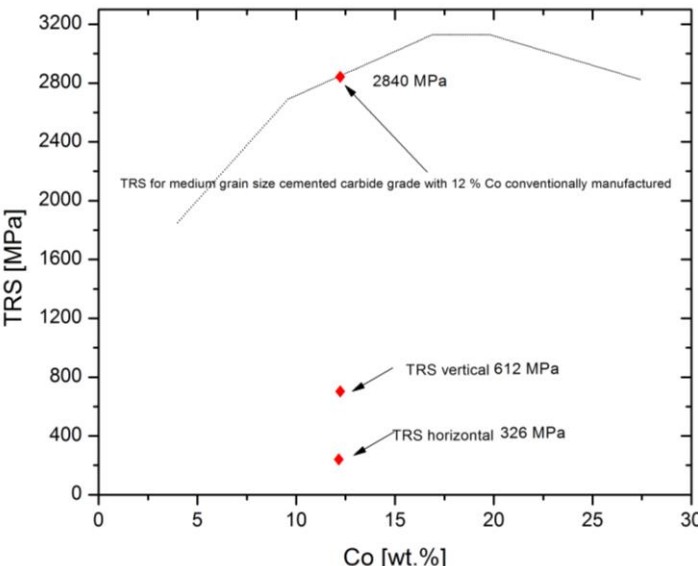

**Figure 4.** Mechanical strength of WC-12Co compared with the data in the literature regarding typical cemented carbides produced by the traditional manufacturing route (PM), data from [19,20].

The Vickers hardness test led to an average of 1285 + 167, 52 HV30 due to four measurements. In Figure 5 are shown the results of the hardness test and the indentation marks made on the microstructure of WC-Co specimens'. Even though the average hardness values obtained on the samples processed by the SLS are similar to the values obtained on samples processed by BJ [7] for the medium grain size cemented carbide grade, there is a big difference between the four hardness measurements that have been performed. We consider that the reason behind the wide hardness values range is the porosity within the specimens, although the measured areas do not present visible porosity on the surface where the indentation has been performed.

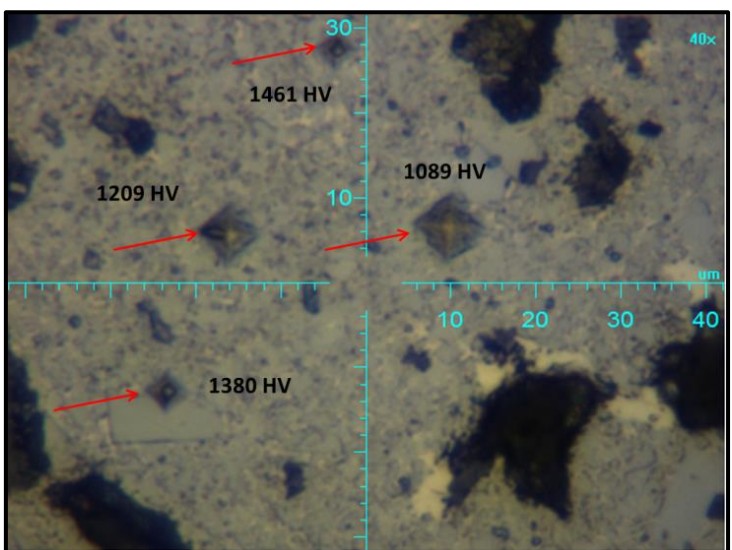

**Figure 5.** Vickers hardness test performed with 0.3 kgf on the WC-12Co specimens processed via SLS and Sinter-HIP.

In Figure 6 is presented the evolution of hardness as a function of cobalt content for cemented carbide grades manufactured in the traditional way [20]. The hardness measured on the WC-12Co specimens processed by SLS and sinter-HIP fits in the hardness range at the border of the cemented carbides with medium grain size (1.2–2.0 μm) and

medium coarse size (2.1–3.4 μm) with 12% Co, (Figure 6), it is comparable with the hardness of the typicall cemented carbides grades as a function of the cobalt content for cemented carbides manufactured by using the conventional way with the same WC grain size [18]. This shows that the SLS followed by Sinter-HIP can potentially obtain cemented carbide parts usable for specific applications, where the need for a given hardness (wear resistance) prevails over the need for high strength.

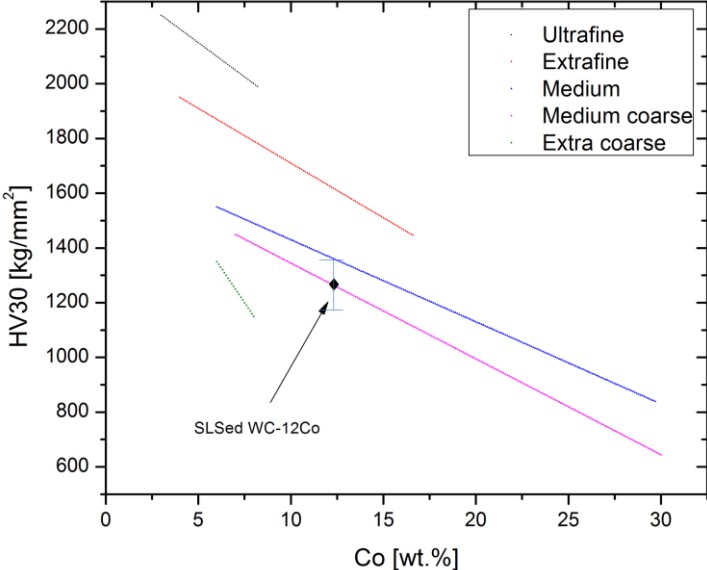

**Figure 6.** Vickers hardness of WC-12C of samples processed by SLS and Sinter-HIP versus the typical hardness of different grades of cemented carbides produced by Powder Metallurgy, data from [20,21].

In SLS, the printing direction can have a significant influence on the quality of the final product. When printing in different directions, the object's orientation can affect the final product's strength, surface finish, and overall quality. The printing direction can also affect the strength and mechanical properties of the final product. The fracture surface, resulting after the bending test, was examined by SEM for the broken TRS specimens built horizontally and vertically, (Figure 7). Regardless of the magnification, the porosity can be seen throughout the entire surface of the fractured samples, a fact in good agreement with the data in the literature [22,23]. Porosity can have a significant impact on the properties of cemented carbides, and it can reduce the strength, toughness, and wear resistance of the material. Porosity can also act as a stress concentrator, leading to premature failure of the material under load. The source of the high porosity is the high amount of organic binder used in the SLS process (20 wt.%) and the residual voids left after the printing process due to hollow WC-Co particles. For the TRS samples built horizontally, (Figure 7b,c), the shape of the former granules of the agglomerates have been flattened, while the samples built vertically have maintained their shape, (Figure 7e,f).

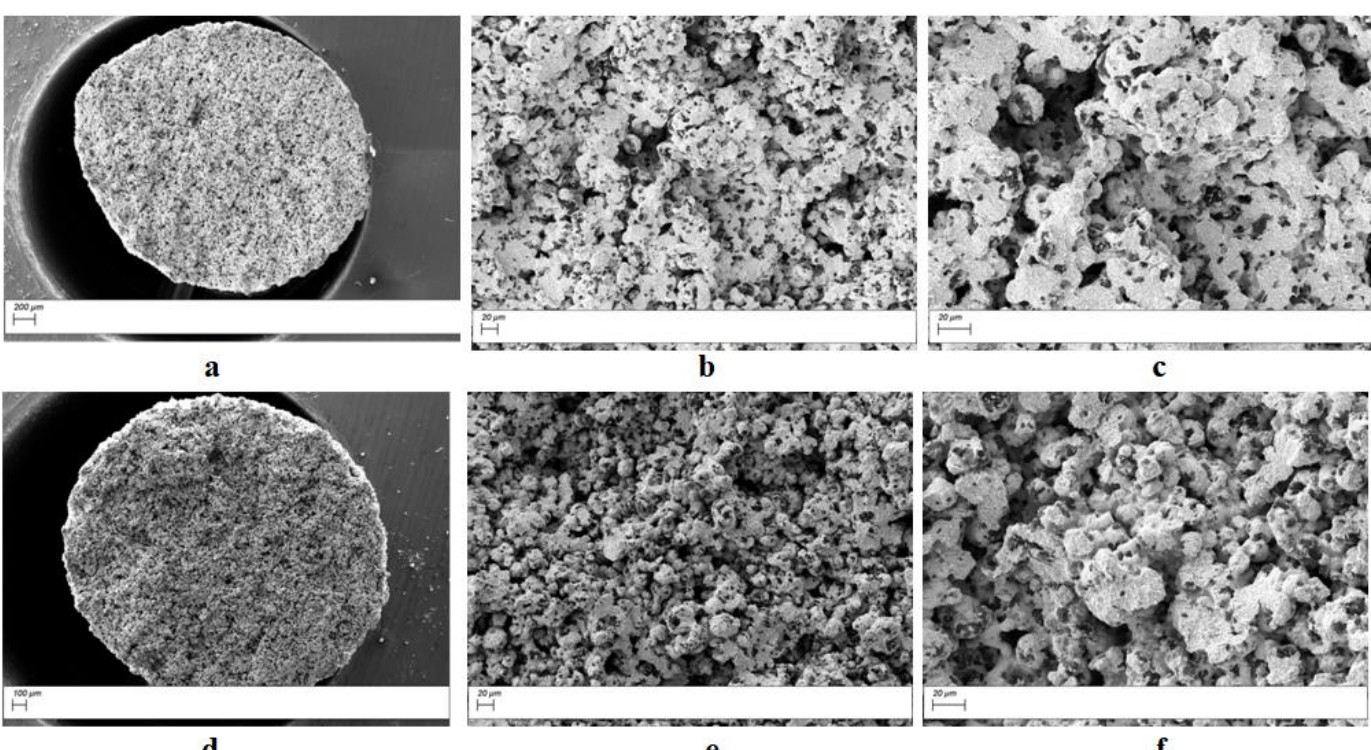

**Figure 7.** SEM images of the TRS fracture surface of WC-12Co samples built horizontally: (**a**) low magnification, (**b**) average magnification and (**c**) high magnification; and WC-12Co samples built vertically: (**d**) low magnification, (**e**) average magnification and (**f**) high magnification.

A horizontal approach used in printing the TRS samples has provided a much higher packing density of the particles, easing the diffusion processes and other densification mechanisms to be active. In contrast, for the samples built vertically, forming the liquid phase and rearranging particles constituted the primary mechanisms by which the sintering necks were created.

There are several ways to have the density increased after the sintering process. One way would be to use a unique design WC-Co powder, which imposes high bulk density because the granules are pre-sintered and sintered several times to remove the inner porosity within the granules [24,25]. In this way, the packing density of the powder bed can be enhanced, therefore, the green and sintered density. On the other hand, higher amounts of cobalt can be used to increase the density through the rearrangement mechanism during the liquid phase sintering [26]. Randal M German [27] stated that if 30% vol of liquid phase is formed during the sintering process, fully densification can occur only through rearrangement. Another way would be to decrease the amount of organic bindery obtaining WC-Co-Organic binder powders through spray drying so that each granule contains organic binder well distributed throughout the whole volume [28].

The failure of WC-12Co cemented carbides is a matter of the sintered particle resistance under various solicitations in working conditions. The machine learning algorithms based on a neural network approach modelled and investigated such materials' behaviour. It was found that reducing the grain size improves the wear resistance, and the high-temperature crack resistance might be improved by adding some rare earth, such as La [29,30].

## 4. Conclusions

This study was carried out to assess the mechanical properties of indirect SLS 3D printed (SLS) WC-Co/PA12 composites. The sintered samples displayed relative densities up to 66%, while the shrinkage of the cube-shape samples has been up to 43%. The microstructure contained WC grains within the 1.4–2.0 µm range, with only large WC grains

isolated. The TRS test has shown a low mechanical strength, up to 612 MPa, for the samples vertically built, while for the samples built horizontally, the mechanical strength is even lower due to the residual porosity after sinter-HIP. The Vickers hardness test led to an average hardness of 1285 HV30 at the border of cemented carbides medium and medium coarse grades traditionally manufactured. The SEM image of the fracture surfaces of the TRS samples has shown the high residual porosity found throughout the entire volume. The obtained results show, on the one hand, the limitations of the process involving SLS for the manufacturing of WC-12Co parts and, on the other, the potential to develop the method for producing certain cemented carbide parts, with complex shapes, in small production scales.

**Author Contributions:** Conceptualisation, O.-D.J. and C.-O.P.; methodology, O.-D.J. and H.-F.C.; software, O.-D.J.; validation, C.-O.P., H.-F.C., and N.B.; formal analysis, R.-V.G.; investigation, R.-V.G.; resources, O.-D.J.; data curation, C.-O.P.; writing—original draft preparation, O.-D.J.; writing—review and editing, O.-D.J. and C.-O.P.; visualisation, N.B.; supervision, C.-O.P. and H.-F.C.; project administration, H.-F.C.; funding acquisition, H.-F.C. and R.-V.G. All authors have read and agreed to the published version of the manuscript.

**Funding:** This research was financially supported by the Project "Entrepreneurial competences and excellence research in doctoral and postdoctoral programs-ANTREDOC", a project co-funded by the European Social Fund financing agreement no. 56437/24.07.2019.585. This work was supported by the european development Fund and the Romanian Government through the Competitiveness Operational Programme 2014–2020, project id P 34 466, MySMiS code 121349, contract no. 5/05.06.2018. The generous support of the Gühring company in making this work possible is highly acknowledged. This work was made possible by the generous support of the Gühring Company.

**Data Availability Statement:** Data are available on request from the correspondent author.

**Conflicts of Interest:** The authors declare no conflict of interest.

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
