# Peer review of "The Assessment of the Transversal Rupture Strength (TRS) and Hardness of WC-Co Specimens Made via Additive Manufacturing and Sinter-HIP"

_metals, doi:10.3390/met13061051_

Round 1
Reviewer 1 Report
The work is interesting and well presented. A correction and a few improvements are however necessary for publication.
On page 6, line 207-209, one reads that the fracture strength in three point bending is computed by dividing the peak load by the specimen cross section, which is contrary to basic beam theory ! This should be corrected (and the data updated accordingly)
The number of fracture tests and the scatter of the results should be specified, and error bars added on Fig. 4
Scatter bars should also be added on Fig. 6
The reason which sintering in inert gases inhibits grain growth should be explained
overall, the English is correct, with just a few corrections needed here and there (for example "closed porosity" instead of "close prorosity", "cobalt rich areas" instead of "rich cobalt areas")
Reviewer 2 Report
In this article, the transversal rupture strength and hardness of WC-Co specimen made via Additive Manufacturing and Sinter-HIP is assessed.
1. Too many Abbreviations and Acronyms are used and it affects readability. Expanded for of all these abbreviations should be given. Eg. HIP
2. The authors claim, WC-Co composite was never made before via SLS and sinter-HIP. However, the article lacks sufficient justification and well framed objectives.
3. The results section should be divided into subsections with subtitles to enhance the readability of the article.
4. Results should be compared with WC-Co specimen developed through other methods by previous works.
5. What about the processing conditions of sintering? Whether they are optimized?
6. High residual porosity is stated as the reason for discrepancies. However, counter measures to avoid this condition should have been discussed and practiced.
Reviewer 3 Report
Please explain the sense of producing a material whose TRS for vertical arrangement is about 80% and for horizontal about 90% compared to conventional material.
No indication of deformation in the TRS test
What is the point of testing the hardness of which the imprints are much smaller than the pores. Normally, the gaps between the prints should be 3 of their diagonals apart.
Author Response
Reviewer 3.
- Q: Please explain the sense of producing a material whose TRS for vertical arrangement is about 80% and for horizontal about 90% compared to conventional material.
A: The purpose of the current research was not to produce parts by SLS and Sinter HIP with a certain application (eg. Cutting tools) due to the fact that using this technology, it is not possible to produce parts fully densified. The idea behind this research is to investigate the use of SLS + the conventional processing way of cemented carbides (SLS) to process WC-Co powder. In this way we could study the capability of the printing process (SLS) together with the conventional process (Sinter-HIP) and to assess how far or how far we are standing in contrast with the cemented carbide grades produce through the conventional process.
- Q: No indication of deformation in the TRS test
A: TRS or the 3-point bending test it is used to assess the mechanical strength of brittle materials ( eg. Ceramics), therefore no major deformation occurs during the test. In the case of cemented carbides, they behave like brittle materials and TRS is a common method to assess their mechanical strength.
- Q: What is the point of testing the hardness of which the imprints are much smaller than the pores. Normally, the gaps between the prints should be 3 of their diagonals apart.
The hardness test performed on the samples printed and sintered has been done by Vickers microhardness. It was done in relatively fully densified to assess the influence of process itself upon the base materials used in this study are ready to press WC-Co and high amount of polyamide powder. The gap between the indentation marks is more the 3 of their diagonals apart. One of the indentation marks has been made on a big WC grain as you can notice. Again, the study was to assess the hardness of the SLS+Sinter HIP samples in contrast with the conventional WC-Co grade to see how far or how close we are regarding mechanical properties.

Round 2
Reviewer 2 Report
Authors have carried out most of the given suggestions.
Some grammatical errors are found in the text. Some editing with the help of a native speaker will be helpful.
Author Response
The manuscript has undergone major language changes done by an english native speaker and a professional software. Please see the manuscript.
Reviewer 3 Report
I accept the answers
Author Response
Thank you.